# Effect of Essential Oil from *Lippia origanoides* on the Transcriptional Expression of Genes Related to Quorum Sensing, Biofilm Formation, and Virulence of *Escherichia coli* and *Staphylococcus aureus*

**DOI:** 10.3390/antibiotics12050845

**Published:** 2023-05-03

**Authors:** Andrés Martínez, Elena E. Stashenko, Rodrigo Torres Sáez, German Zafra, Claudia Ortiz

**Affiliations:** 1Grupo de Investigación en Bioquímica y Microbiología (GIBIM), Escuela de Microbiología, Facultad de Salud, Universidad Industrial de Santander, Bucaramanga 680002, Colombia; 2Escuela de Química, Centro de Cromatografía y Espectrometría de Masas (CROM-MASS), Universidad Industrial de Santander, Bucaramanga 680002, Colombia

**Keywords:** biofilm, essential oils, *Lippia origanoides*, gene expression analyses, *E. coli*, *S. aureus*

## Abstract

Microbial infections resistant to conventional antibiotics constitute one of the most important causes of mortality in the world. In some bacterial species, such as *Escherichia coli* and *Staphylococcus aureus* pathogens, biofilm formation can favor their antimicrobial resistance. These biofilm-forming bacteria produce a compact and protective matrix, allowing their adherence and colonization to different surfaces, and contributing to resistance, recurrence, and chronicity of the infections. Therefore, different therapeutic alternatives have been investigated to interrupt both cellular communication routes and biofilm formation. Among these, essential oils (EO) from *Lippia origanoides* thymol-carvacrol II chemotype (LOTC II) plants have demonstrated biological activity against different biofilm-forming pathogenic bacteria. In this work, we determined the effect of LOTC II EO on the expression of genes associated with quorum sensing (QS) communication, biofilm formation, and virulence of *E. coli* ATCC 25922 and *S. aureus* ATCC 29213. This EO was found to have high efficacy against biofilm formation, decreasing—by negative regulation—the expression of genes involved in motility (*fimH*), adherence and cellular aggregation (*csgD*), and exopolysaccharide production (*pgaC*) in *E. coli*. In addition, this effect was also determined in *S. aureus* where the *L. origanoides* EO diminished the expression of genes involved in QS communication (*agrA*), production of exopolysaccharides by PIA/PNG (*icaA*), synthesis of alpha hemolysin (*hla*), transcriptional regulators of the production of extracellular toxins (RNA III), QS and biofilm formation transcriptional regulators (*sarA*) and global regulators of biofilm formation (*rbf* and *aur*). Positive regulation was observed on the expression of genes encoding inhibitors of biofilm formation (e.g., *sdiA* and *ariR*). These findings suggest that LOTCII EO can affect biological pathways associated with QS communication, biofilm formation, and virulence of *E. coli* and *S. aureus* at subinhibitory concentrations and could be a promising candidate as a natural antibacterial alternative to conventional antibiotics.

## 1. Introduction

Antimicrobial resistance (AMR) to conventional antibiotics is a serious public health problem directly causing an estimated 1.3 million deaths per year around the world [1,2]. These infections can be considered emergent diseases because of their potential to affect human beings and the limitations of the therapeutic treatments for them around the world [3,4]. Among different antimicrobial-resistant microorganisms, *E. coli* and *S. aureus* are the most prevalent pathogens, mainly because they can form biofilms. Biofilm-forming bacteria can produce a compact and protective matrix allowing them to adhere to different surfaces such as medical devices and cellular tissues. Microbial growth of these pathogens generally contributes to the chronicity of the infection and its recurrence, especially in both implants and medical devices [5,6,7].

*S. aureus* is the main microorganism causing nosocomial and community-acquired bacterial infections [8]. Methicillin-resistant and multidrug-resistant *Staphylococcus aureus* (MRSA) strains are becoming a serious threat to global public health, which stimulates the search for new antimicrobial agents [9]. During biofilm formation, *S. aureus* can produce diverse virulence factors, including hemolytic toxins, enterotoxins, and proteolytic enzymes, among others. One important virulence factor is the pore-forming toxin alpha-hemolysin (*hla*) [10]. This *hla* has a strong hemolytic effect on red blood cells from different mammals and plays an important role in biofilm formation in staphylococcal infections [11]. In addition, QS-agr and global regulators such as *sarA*, aur and *rbf* coordinately control the colonization, adhesion, and exopolysaccharide formation in *S. aureus* infections. The formation of different types of polysaccharide intercellular adhesin (PIA) and Poly- β(1-6)-N-acetylglucosamine (PNAG)-dependent exopolysaccharides or extracellular proteases allows *S. aureus* to invade any type of biotic and abiotic surface. This biological property converts *S. aureus* into one of the most important microorganisms associated with nosocomial infections on medical devices [12,13].

On the other hand, *Escherichia coli* is a recognized pathogen causing different important intestinal and extraintestinal infections [14]. Some *E. coli* strains have been implicated in sporadic cases and outbreaks of enterohemorrhagic diarrhea throughout the world [15] and are one of the most common multidrug-resistant strains of urinary tract infection in Latin America [16,17]. *E. coli* possesses essential virulence factors for adhesion to epithelial cells and cellular aggregation, which drives biofilm formation in cell infections [18,19]. In pathogenic strains of *E. coli*, the *LuxS* gene is responsible for QS regulation, an important tool in the regulation of gene expression of some virulence factors and bacterial motility of these bacterial strains [20]. During *E. coli* biofilm formation, gene regulation of bacterial mechanisms such as motility, adhesion, and cellular aggregation is an essential issue [21]. For instance, motility is influenced by the regulation of flagella and pili that facilitate cell–surface interaction and cellular aggregation, curli and fimbriae syntheses that enable cellular communication and exopolysaccharide formation, and therefore promote an irreversible interaction between bacteria and cell surfaces [22]. Each of these bacterial features promotes biofilm formation, chronicity, and antibiotic resistance. Among the different treatments explored to combat AMR, essential oils (EOs) have emerged as promising mixtures against infections caused by antibiotic-resistant microorganisms. EOs are secondary metabolites with antimicrobial properties, generally acting on the cell membranes and therefore affecting the cellular structures of microorganisms, which facilitates their cytotoxic and therapeutic properties [23,24,25]. Recently, EO from *Lippia origanoides* (Verbenaceae family), mainly composed of phenolic monoterpenes, has been proven to have high antimicrobial activity against different pathogenic microorganisms [16,26,27]. These findings provide further evidence of the potential of the *L. origanoides* EO as an antimicrobial agent against infections caused by *S. aureus* and *E. coli*. Therefore, this work aimed to study, via RT-qPCR analyses, the effect of the *L. origanoides* EO from on the expression of genes related to QS communication, biofilm formation, and virulence of pathogenic strains of *E. coli* ATCC 25922 and *S. aureus* ATCC 29213.

## 2. Results

### 2.1. Chemical Composition of the L. origanoides EO

Five major components present in *L. origanoides* EO were identified via GC-MS analyses [28] (See Table 1). The percentage of major biomarkers are as follows: oxygenated compounds 51.5%, sesquiterpenes 6.3% and monoterpenes 6.4%, respectively. Among them, the major biomarkers were thymol (32.7%) and carvacrol (18.8%), which have been previously proven as promising antimicrobial compounds against antibiotic-resistant bacteria [29].

### 2.2. Antimicrobial Activity of the LOTC II EO on Biofilm Formation of E. coli and S. aureus

The anti-biofilm effect of the LOTC II EO on *E. coli* ATCC 25922 and *S. aureus* ATCC 29213 cultures is shown in Table 2 and Figure 1. A high inhibitory effect on biofilm formation was observed with bacterial cultures treated with the LOTC II EO. Inhibition of biofilm formation by 76% and 71% for *E. coli* and *S. aureus*, respectively, was determined for this EO at a CMIB of 0.40 mg/mL.

### 2.3. Obtaining Cell Biomass from Treated and Untreated Bacterial Biofilms with LOTC II EO in Bioreactors of 50 mL

Scale-up of biofilm cultures was performed to obtain a large amount of cell biomass. Initially, inhibition kinetics were performed on a subinhibitory concentration of the LOTC II EO on planktonic and sessile cells in the bioreactors. This was carried out to assess cell culture conditions in the bioreactor. Therefore, it was corroborated that antibacterial and antibiofilm activity determinations of the LOTC II EO were not altered at the culture conditions in the bioreactor at volumes of 50 mL. Figure 2 shows the cell inhibition kinetics obtained with *E. coli* cultures treated with EO.

### 2.4. Total RNA Extraction and cDNA Synthesis

Extraction of total RNA from biofilm samples treated and untreated with the subinhibitory concentration of EO from LOTC II was performed at 24 h of incubation time. Table 3 shows the RNA properties of each biofilm samples. All RNA exhibited A260/280 ratio of ~2.0, which indicated that the RNA samples had adequate purity and yield; therefore, these samples could be used for amplification experiments by RT-qPCR.

### 2.5. EO Effect on Swimming Motility of E. coli ATCC 25922

All the evaluated genes are directly involved in the QS regulation system, biofilm formation, and virulence of both *E. coli* and *S. aureus*. The specificity of the synthesized primers was evaluated via agarose gel electrophoresis and real-time PCR. Figure 3 shows agarose gel electrophoresis of evaluated genes from *E. coli* and *S. aureus*. All amplifications showed a single amplification product.

### 2.6. Differential Expression Analysis of Genes Related to Quorum Sensing, Biofilm Formation, and Virulence

Our results showed significant differences between treated and untreated cultures with the EO for all evaluated microorganisms (See Figure 4 and Figure 5), with higher changes observed for *E. coli*. Both positive and negative regulation of genes related to quorum sensing and biofilm formation were observed. In *E. coli*, the expression of genes related to motility and production of biofilm exopolysaccharide was mainly affected, while in *S. aureus*, the expression of genes related to the global regulation of exopolysaccharide production and cell survival was mainly modified.

## 3. Discussion

Biofilm formation in *S. aureus* is a multifactorial process influenced by different biological processes and factors, with PIA being one of the most important. Although different candidate polysaccharides have been postulated to be determinants of biofilm formation, PIA, a PNAG, is the main exopolysaccharide component of the staphylococcal biofilm matrix and is linked to irreversible adhesion of *S. aureus* [30]. Enzymes for PIA/PNAG synthesis are encoded by the *icaADBC* operon, and any mutation of this gene operon causes a decreased capacity for biofilm formation [31]. Within the *icaADBC* operon, the *icaA* and *icaD* genes are directly related to PNAG synthesis and cellular multilayer clustering, whereas the *icaB* and *icaC* genes encode for a protein involved in matrix exopolysaccharide stability and a protein receptor from polysaccharides, respectively. In addition, the activation of this operon is influenced by the negative regulation of the *icaR* gene and the activation of the agr system [32,33]. In this study, we found a negative regulation in the expression of the *icaA* gene in *S. aureus* with no significant changes in the expression of the *icaD* gene. The regulation of these genes is involved in the synthesis of adhesins and exopolysaccharide exportation [30]. Moreover, gene transcription of proteins from *icaADBC* is under positive global regulation of the *sarA* transcription regulator [34]. Negative regulation of *sarA* gene expression was observed in cultures of both *E. coli* and *S. aureus* treated with the EO from LOTC II. The *sarA* locus encodes a DNA-binding protein required in some conditions for microbial growth associated with biofilm formation. Previous studies [35,36] proved that a *sarA* mutation causes a decrease in biofilm formation and a diminished transcription of genes of the *icaADBC* operon and PIA/PNAG synthesis. Moreover, these studies showed that the negative regulation of the *icaA* and *icaD* genes, even only the *icaA* gene, caused a significant reduction in biofilm formation in *S. aureus* [37].

On the other hand, the expression of the *agrA* gene was decreased by the effect of the EO from LOTC II in both biofilm and planktonic cells of *S. aureus*. This accessory gene regulator (*agr*) is important in the regulation of the QS mechanism and pathways associated with the synthesis of the exopolysaccharide matrix. In the development of *S. aureus* biofilms, some cell surface proteins play an important role in the adhesion of bacterial cells to host cells and surfaces; among these, microbial surface components that recognize adhesive matrix molecules (MSCRAMMs), mediate the adhesion of microbes to components of the extracellular matrix of the host. On staphylococci, MSCRAMMs often have multiple ligands, and their production is an essential step in the formation, development, and maturation of biofilms [38,39]. The MSCRAMM synthesis is influenced by the *agr* and the staphylococcal accessory regulator (*sarA*). These regulatory elements play opposing roles in *S. aureus* biofilms formation because mutation of *agr* results in increased biofilm formation and decreased antibiotic susceptibility, while mutation of *sarA* has the opposite effect [35,40,41].

The *agr* locus encodes a two-component QS system that modulates the synthesis of a transcriptional regulator (*RNA III*) and the autoregulation of the agr system. The LOTC II EO also affected the gene expression of the *RNA III* gene, significantly decreasing its transcription in the biofilm formation of *S. aureus*. *RNA III* is an important transcriptional regulator of biofilm formation in *S. aureus* and is responsible for the posttranscriptional regulation of several virulence factors that mediate changes in the expression of cell surface-related proteins and extracellular toxins such as alpha-hemolysin (*hla*) and delta-hemolysin (*hld*) [42]. Caiazza et al., (2003) proved that *hla* synthesis was necessary for biofilm formation by an activation mechanism of adhesive proteins [43,44]. In addition, negative regulation of the *hla* gene by the effect of the LOTC II EO suggests that this EO affects not only alpha-hemolysin synthesis but also QS signal recognition proteins, causing inhibition on the expression of transcriptional regulators, toxin production, and biofilm formation. The *agrA* gene also encodes an essential protein in QS signal recognition and acts as a transcriptional regulator of different bacterial features from *S. aureus*. Previous studies showed that bacterial strains expressing agr genes at high levels had a decreased capacity for biofilm formation, that is, inactivation of the QS system in *S. aureus* would be necessary for biofilm reinforcement [45,46]. However, agrC and agrA comprise a classic two-component signal transduction system, where agrC bound to a ligand activates a DNA-binding response regulator agrA. In this case, active dimers of agrA are bound to an intergenic region of agr and positively regulate the expression of the two operons. Moreover, agrA independently regulates the expression of cytolytic phenol soluble modulins (PSMs) and several genes related to cell metabolism [47].

Biofilm formation in *S. aureus* is not only mediated by PIA/PNAG synthesis. It is possible that biofilm formation is dependent on the *icaADBC* operon biofilm formation. These biofilms are associated with the biofilm-associated protein (Bap), a surface protein implicated in the biofilm formation of *S. aureus* strains isolated from chronic infections. Regulation of Bap-dependent biofilms is influenced by global regulators such as *rbf* [48]. In this study, negative regulation of the expression of *rbf* and aur genes in response to the LOTC II EO was observed. These genes are related to protein exopolysaccharide production and extracellular proteases. Additionally, it has been proposed that *rbf* is involved in the positive regulation of important proteins of biofilm formation [49]. Lim et al., (2004) observed that the insertion of the *rbf* gene in *S. aureus* altered biofilm formation on polystyrene and glass surfaces. Nevertheless, this mutant was not affected in its primary adhesive step, which suggests that *rbf* inactivation affects cell aggregation but not cell adhesion and can regulate *ica* genes via an independent pathway [50]. These findings were previously observed by Cue et al., (2009), who found that *Rbf* represses *icaR* transcription with a concomitant increase in *icaADBC* gene expression and enhanced PNAG and biofilm formation [51]. Thus, inhibition of *S. aureus* biofilm formation by the LOTC II EO would affect both PIA and protein exopolysaccharide biosynthesis.

The first stages in biofilm formation in *E. coli* require the synthesis of different structures from bacterial surfaces that allow irreversible attachment to cell surfaces [52]. In this sense, adhesive organelles such as curli fimbriae, encoded by the *csg* operon, and type I fimbriae, encoded by *fim* genes, are especially important. On the other hand, cell motility is a mediator of cell–cell interactions and acts as a determining factor of biofilm architecture [53]. Additionally, motility and synthesis of fimbriae-flagellum would be a key factor in the development of QS communication and biofilm formation in *E. coli*, since once the bacteria cells are irreversibly bound to the surface, coproduction of polysaccharides and curli is necessary for biofilm development [52]. Genes for curli synthesis are organized into two divergent operons: *csgBA*, encoding structural components, and *csgDEFG*-encoding proteins for assembling and transporting of curli. Gene expression of these two curli operons is under the control of the csgD protein. In this study, we found a negative regulation of the expression of the *csgD* gene in *S. aureus* caused by the LOTC II EO, which is an important finding since the csgD protein modulates the expression of a set of genes responsible for the adaptation of cell physiology to the biofilm state [54,55].

Type I fimbriae, known as pili, are commonly used as adherence structures to resist shear stress. In *E. coli*, one of the most important proteins constituting the type I fimbriae is the *fimH* protein, a highly conserved adhesive subunit responsible for structure maintenance. The *fimH* domain is responsible for the adherence process, a main step in the colonization of biofilm formation of *E. coli* [56]. Zuberi et al., (2017), proved that deletion of the *fimH* gene blocks the synthesis of the *fimH* subunit of fimbriae from *E. coli*, significantly reducing its ability for biofilm formation [57]. We found that the LOTC II EO caused a negative regulation of the expression of the *fimH* gene in *E. coli* biofilms, suggesting a loss of motility and an effect on curli proteins, with the concomitant inhibition of biofilm formation because it is affected cell–surface and cell–cell interactions, and blocked cell aggregation.

We also found a negative regulation in the expression of the *pgaC* gene of *E. coli* in response to the LOTC II EO; this gene is involved in exopolysaccharide synthesis pathways and exportation. In biofilm maintenance of *E. coli*, the production of the linear homopolymer poly-Beta-1,6-N-acetyl glucosamine (PGA) is important because this polymer acts as an adhesin, giving shape and stability to bacterial biofilms. PGA synthesis requires the expression of the *pgaABDC* operon, which is necessary for the maturation of the biofilm [58]. In addition, the *pgaC* protein is an essential glucosyltransferase for PGA production. It has been proven that deletion of the *pgaC* gene blocks PGA synthesis, inhibiting the maintenance and formation of biofilms [59]. Consequently, the negative regulation of *pgaC* gene expression caused by the LOTC II EO affected exopolysaccharide production, significantly inhibiting biofilm formation and cell aggregation, which was clearly observed in SEM analyses [27].

In contrast, positive regulation of *ariR*, an important gene involved in resistance pathways to environmental changes, was observed in response to LOTC II EO. During the sessile growth of *E. coli*, ariR is an important protein because it is a global regulator that is upregulated by cytoplasmic pH stress and therefore allows *E. coli* to resist acidic conditions. Transcriptomic studies have identified that this protein plays an important role in the colonization of *E. coli* in the digestive system and is involved in cell communication and biofilm development [60,61]. Phenotype studies showed that *ariR* represses biofilm formation in under stress environments, decreasing cell motility and protecting the bacterial cells against acidic conditions. These data are potentially mediated through AI-2 signal interactions (*luxS*) and indole, which suggests that *ariR* is a nonspecific transcriptional regulator [61]. Thus, overexpression of the *ariR* gene facilitates the resistance of *E. coli* to environmental pH changes, which would be caused by the effect of the LOTC II EO on bacterial cells, with the consequent inhibition of motility and biofilm formation.

Moreover, the LOTC II EO positively regulated the expression of *LuxS* and *gseC* genes in *E. coli*. These genes are associated with cell communication and are important in biofilm formation. The AI-2 is the main QS communication system in *E. coli* since autoinducer-2 (furanosyl borate diester) synthesis is regulated by the *LuxS* protein [62]. AI-2 system promotes biofilm formation and changes its structure when it stimulates flagellar motility through the QS motility regulator *MqsR*. In addition, this *MqsR* regulator acts through the two-component *QseBC* motility regulator system. Thus, the two-component *QseBC* motility regulator system would transcriptionally affect cell motility gene expression [63,64]. Additionally, Yang et al., (2014) proved that the *QseC* histidine kinase sensor, a *QseB* response protein regulator, plays an important role in an additional cell communication system for biofilm formation mediated by the epinephrine–norepinephrine (EPI-NE) recognition process in *E. coli* [65]. Therefore, *E. coli* cells could regulate their motility mechanisms through the regulation of the *gseC* gene; however, this hypothesis should be confirmed by studies on changes in *gseB* gene expression.

On the other hand, in this study, a positive regulation in the expression of the *SdiA* gene, a transcriptional regulator related to QS communication, was observed in *E. coli* biofilm cultures treated with the LOTC II EO. *E. coli* encodes a transcription-activating protein associated with QS communication, a homolog receptor of *LuxR* known as suppressor of division inhibitor (*SdiA*). Although *E. coli* is not able to synthesize N-acyl homoserine lactone (AHL) molecules, *SdiA* can recognize autoinducer molecules produced by other bacteria. *E. coli* uses *SdiA* proteins to reduce biofilm formation by recognizing QS and indole signals. Previous studies have proven that *SdiA* reduces biofilm formation by repressing genes related to curli and indole pathway synthesis. These results suggest that *E. coli* can regulate *sdiA* expression to decrease biofilm formation by altering signal sensors [66]. Culler et al., (2018) showed that *SidA* is active and functional in the presence and absence of AHL molecules. Moreover, *SdiA* can sense different environmental conditions, such as osmolarity and temperature, allowing *E. coli* to regulate the stress response system and survive in the infected host or in the environment [67]. Kim et al., (2014) proved the interaction of *SdiA* and the cell division promotor *ftsQP2* as a response to stress in the absence of inducing molecules [68]. Therefore, the positive regulation of *SdiA* expression caused by the LOTC II EO would significantly affect the motility mechanism and biofilm formation in *E. coli*.

## 4. Materials and Methods

### 4.1. Materials

#### Bacterial Strains and Plant Material

*Escherichia coli* ATCC 25922 and *Staphylococcus aureus* ATCC 29214 were purchased commercially from ATCC by Grupo de Investigación en Bioquímica y Mcirobiologia (GIBIM). *Lippia origanoides* chemotype thymol-carvacrol II plants were harvested from experimental plots located at the Agroindustrial Pilot Complex of CENIVAM (National Research Center for the Agro-Industrialization of Tropical Medicinal Aromatic Plants), at Universidad Industrial de Santander (Bucaramanga, Colombia). The taxonomic characterization of the plants was carried out at the Institute of Natural Sciences of the Universidad Nacional de Colombia (Bogotá, Colombia) and they were identified at the species level [28].

### 4.2. Essential Oil Distillation and Analysis

EO from *Lippia origanoides* thymol-carvacrol II plants was extracted via microwave-assisted hydrodistillation (MWHD) and characterized using gas chromatography coupled to mass spectrometry (GC/MS) [69].

### 4.3. Determination of the Minimum Inhibitory Concentration of the LOTC II EO on E. coli and S. aureus Biofilm Formation

Inhibition of biofilm formation of *E. coli* and *S. aureus* cultures by EO from LOTC II was determined as described by Martínez et al., (2021), with some modifications [27]. Briefly, sterile flat-bottom polystyrene (PS) 96 microtiter plate wells were used for biofilm formation. Cultures were grown overnight in 3 mL of tryptic soy broth (TSB) with 2% *w*/*v* glucose diluted (1/100) in growth medium to 5.8 × 10^5^ (CFU/mL) for *S. aureus*, whereas we used Luria Bertani (LB) medium diluted (1/10) in growth medium to 6 × 10^6^ (CFU/mL) for *E. coli*. One hundred microliters of the respective growth culture medium were transferred into the microplate in the presence of 100 µL subinhibitory concentrations of the LOTC II EO. We used 100 µL of bacterial inoculum and 100 µL of peptone water as biofilm formation controls. Microplates were incubated at 27 °C for 24 h. The formed biofilms were then washed three times with sterile phosphate-buffered saline (PBS pH 7.2) to remove free-floating planktonic bacteria. The biofilm formed by adherent sessile organisms in the microplates was stained with 0.45 *w*/*v* crystal violet. All the experiments were performed in triplicate. The inhibition percentage was determined according to the following equation:Inhibition (%) = [(OD negative control − ODEO-treated)/OD negative control] × 100

### 4.4. Obtaining Biomass from Biofilm Treated and Untreated with the LOTC II EO II Bioreactors

Cell biomass from biofilms treated and untreated with the EO from LOTC II was obtained in aerated and stirred 50 mL glass bioreactors, using frosted glass coupons (~15 cm × 2 cm) as a support for biofilm formation. Bioreactors containing TSB culture medium with sub-(MIBC)_50_ of 0.375 mg/mL of the LOTC II EO cultures were inoculated with either 10^7^ CFU/mL of *E. coli* in LB culture medium or 10^7^ CFU/mL of *S. aureus*. Negative controls for EO untreated bioreactors were prepared by inoculating 10^7^ CFU/mL for both *E. coli* and *S. aureus* in peptone water medium. All cultures were carried out at 37 °C for 24 h with constant oxygenation. Subsequently, each coupon was washed three times (3×) with PBS buffer (pH 7.2) to remove unattached bacterial cells. Adhered cells and biofilm were physically removed with a spatula and transferred to 50 mL Falcon tubes and dispersed with 50 mL of peptone water. Finally, total RNA was extracted from these bacterial biofilm cells and from planktonic cells treated and untreated with the LOTC II EO. All the extractions were performed in triplicate.

### 4.5. Extraction of Total RNA and Synthesis of cDNA

Total RNA extractions from biofilm treated and untreated with EO were carried out using the PureLink RNA mini kit (Thermo Fisher Scientific, Waltham, MA, USA), according to the manufacturer’s instructions. The concentration and purification of total RNA was spectrophotometrically assessed using an IMPLEN NanoPhotometer NP80 (Thermo Fisher Scientific, Waltham, MA, USA). A 260/280 absorbance ratio was used as an indicator of purity and protein contamination of RNA samples. Subsequently, cDNA synthesis from total RNA was performed with a RevertAidTM H Minus First Strand cDNA kit according to the manufacturer’s instructions (Fermenta, Thermo Fisher Scientific, Madison, WI, USA). All RNA samples were used to obtain cDNA and adjusted at a final concentration of 10 ng/µL.

### 4.6. Primer Design

Primers for the specific genes listed in Table 4 and Table 5 were designed using Primer3 [70], OligoCalc [71] and SnapGene Tool and Viewer software (6.0.2 version). Primer design was performed according to recommended NCBI protocols. Genes were selected based on different biological processes related to biofilm formation, QS communication, and bacterial regulation features such as pathogenicity and virulence. Some genes have even been previously described in the literature.

### 4.7. Analysis of Differential Gene Expression

Quantification of expressed genes was carried out in the CFX96™ Real-time PCR system and software (Bio-Rad, Hercules, CA, USA). RT-qPCR reactions were performed according to the manufacturer’s instructions using a Luna^®^ Universal SYBR green qPCR 2X Master mix (New England Biolabs, Ipswich, MA, USA) in a total volume of 20 μL, containing 10 μL of Luna Universal qPCR 2X Master Mix, 100 ng of cDNA template, 0.25 μM of forward primer, 0.25 μM of reverse primer and sterile nuclease-free water to complete 20 μL. cDNA amplification involved an incubation for initial denaturation at 95 °C for 1 min, followed by 40 cycles of 95 °C for 15 s, and 55–60 °C for 45 s. After 40 cycles, a melting curve was determined using SYBR green fluorescence. Negative controls for gene quantification were performed by omitting the cDNA template from an amplification reaction. Normalization of amplification curves of genes was determined using *S. aureus* (*muc*) and *E. coli* (*rssA*) housekeeping reference genes. Gene expression and quantification of amplification efficiency were carried out using the 2-ΔΔCt method [82].

### 4.8. Data Analysis

All experiments were performed in triplicate and a one-way analysis of variance (ANOVA) was performed to analyze the results among treatments. The significance level in each assay was <0.05%. The assumption of normality and data variances were previously tested using the Shapiro–Wilk and Levene tests, respectively.

## 5. Conclusions

The LOTC II EO caused significant changes in the expression of genes of *E. coli* ATCC 25922 and *S. aureus* ATCC 29213. These genes were related to adhesion mechanisms and cellular motility mechanisms, exopolysaccharide production (PIA/PNAG), environmental processing, two-component systems, ABC transporter membrane proteins, and global regulators of transcription, which could explain the antimicrobial, anti-QS, and anti-biofilm formation effects at subinhibitory concentrations of the LOTC II EO against both of the studied microorganisms. Through correlations of changes in differential gene expression with metabolic pathways, we suggest a probable mechanism of action; on *E. coli* ATCC 25922 and *S. aureus* ATCC 29213. This mechanism is associated with the inhibition of gene expression of important biological processes of bacterial cells such as motility, surface adhesion, cellular aggregation, exopolysaccharide production, and transcriptional regulators of QS communication and biofilm formation. These results could pave the way for new studies aimed at determining possible therapeutic targets and for the development of new antimicrobial compounds.

## Figures and Tables

**Figure 1 antibiotics-12-00845-f001:**
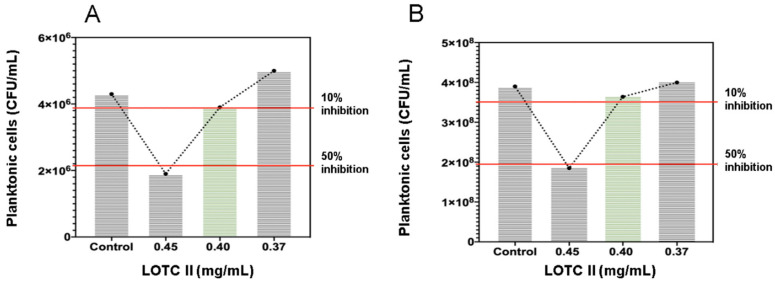
Effect of different subinhibitory concentrations of the LOTC II EO on bacterial planktonic cell cultures. (**A**) *E. coli* ATCC 25922; (**B**) *S. aureus* ATCC 29213.

**Figure 2 antibiotics-12-00845-f002:**
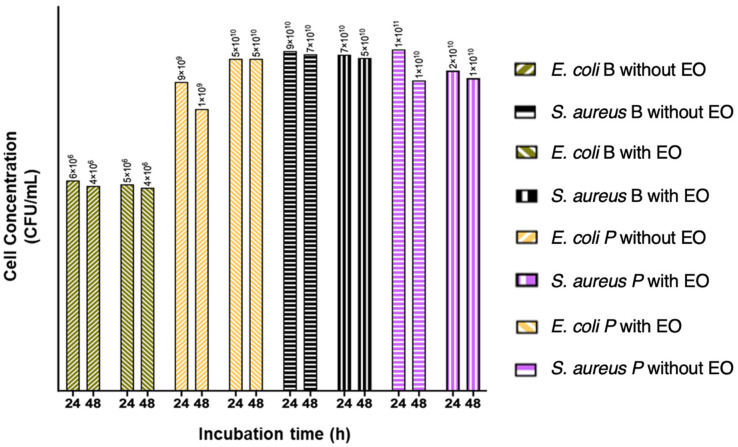
Evaluation of the inhibitory effect of the LOTC II EO at a concentration of 0.37 mg/mL on planktonic and sessile cells in 50 mL bioreactors cultivated with *E. coli* and *S. aureus*.

**Figure 3 antibiotics-12-00845-f003:**
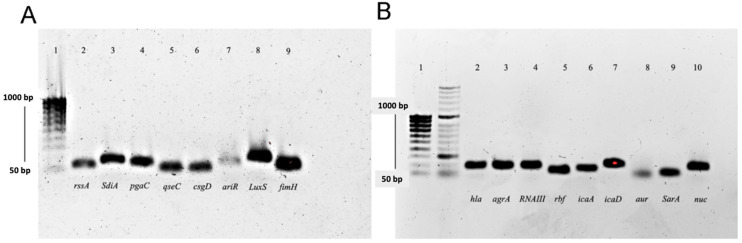
Agarose gel electrophoresis of amplicons of different genes obtained by RT-qPCR of samples from (**A**) *E. coli* ATCC 25922. Lanes: 1. MW markers (50–1000 pb), 2. *rssA*, 3. *SdiA*. 4. *Pgac*, 5. *qseC*, 6. *csgD*, 7. *ariR*, 8. *LuxS*, 9. *fimH*. (**B**) *S. aureus* ATCC 29213. Lanes: 1. MW markers (50–1000 pb), 2. *hla*, 3. *agrA*, 4. *RNAIII*, 5. *rbf*, 6. *icaA*, 7. *icaD*, 8. *aur*, 9. *SarA*, 10. *nuc*.

**Figure 4 antibiotics-12-00845-f004:**
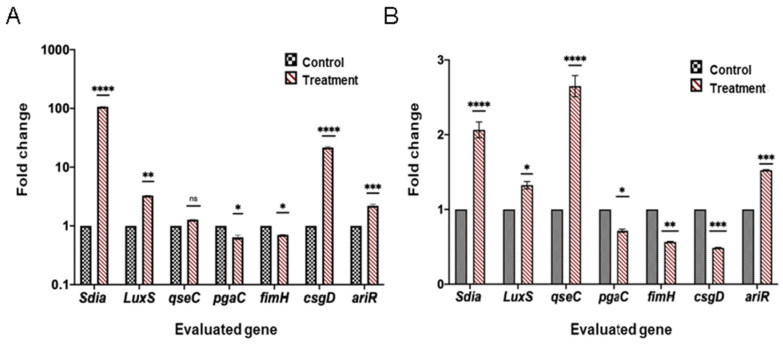
Transcriptional profiles of genes expressed in different cell culture stages of *E. coli* ATCC 25922 treated and untreated with the LOTC II EO. (**A**) Planktonic cells, (**B**) Biofilm. The relative expression of target genes was normalized to the reference genes. All data represent transcriptional levels of genes after EO treatment versus untreated controls at 24 h incubation time. Statistical differences are indicated with asterisks (* *p* ≤ 0.05, ** *p* ≤ 0.02, **** *p* ≤ 0.0001, *** *p* ≤ 0.0002, ns: not significant).

**Figure 5 antibiotics-12-00845-f005:**
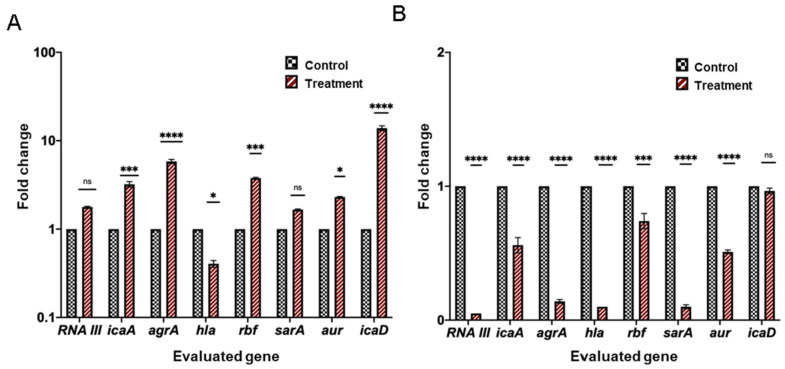
Transcriptional profiles of genes expressed in different cell culture stages of *S. aureus* ATCC 29213 treated and untreated with the LOTC II EO. (**A**) Planktonic cells, (**B**) Biofilm. The relative expression of target genes was normalized to the reference genes. All data represent transcriptional levels of genes after EO treatment versus untreated controls at 24 h incubation time. Statistical differences are indicated with asterisks (* *p* ≤ 0.05, ** *p* ≤ 0.02, **** *p* ≤ 0.0001, *** *p* ≤ 0.0002, ns: not significant).

**Table 1 antibiotics-12-00845-t001:** Five major chemical constituents of the *L. origanoides* EO. The relative amount of each compound is reported as a percentage (%).

Code	Plant Species	Chemotype	Major Components
LOTC II	*Lippia origanoides* (Verbenaceae)	Thymol-carvacrol II	*γ*-Terpinene (5.2%), *p*-cymene (1.1%), thymol (32.7%), carvacrol (18.8%), and *trans*-*β*-caryophyllene (6.4%)

**Table 2 antibiotics-12-00845-t002:** Effect of different subinhibitory concentrations of the LOTC II EO on biofilm formation of *E. coli* and *S. aureus*.

LOTC II (mg/mL)	*E. coli* ATCC 25922			*S. aureus* ATCC 29213		
	Absorbance (OD 595 nm)	Biofilm Formation Inhibition (%)	Planktonic Cell Concentration (CFU/mL)	Absorbance (OD 595 nm)	Biofilm Formation Inhibition (%)	Planktonic Cell Concentration (CFU/mL)
0.37	0.410	24	5.0 × 10^6^	0.730	19	4.10 × 10^8^
0.40	0.134	76	3.9 × 10^6^	0.253	72	3.60 × 10^8^
0.45	0.034	94	1.9 × 10^6^	0.045	95	1.85 × 10^8^
Control	0.540	-	4.3 × 10^6^	0.896	-	3.90 × 10^8^

**Table 3 antibiotics-12-00845-t003:** Evaluation of the concentration and quality of total extracted RNA from treated and untreated samples with the LOTC II EO.

Sample Condition	Concentration (ng/μL)	Absorbance Ratio (260/280)
Planktonic *E. coli* with no-treatment	63	1.99
Planktonic *E. coli* with treatment	58	2.02
Planktonic *E. coli* with no-treatment	49	2.10
*E. coli* biofilm with treatment	43	1.98
Planktonic *E. coli* with no-treatment	120	2.00
Planktonic *S. aureus* with treatment	100	1.98
Planktonic *E. coli* with no-treatment	97	2.01
*S. aureus* biofilm with treatment	80	2.00

**Table 4 antibiotics-12-00845-t004:** List of genes and their respective primers for evaluation of the effect of the EO from LOTC on *E. coli* gene expression during QS communication and biofilm formation.

Gene	Primer	Sequence 5′ to 3′	Product Size (pb)	Tm (°C)	% GC	References
*SdiA*	*SdiA 1* *SdiA 2*	CGGTGCTGAACCCTGAA CGCTGCAACGGGAAAA	177	59.3 60.5	58.8 56.2	(This work)
*LuxS*	*LuxS 1* *LuxS 2*	TGTTGCTGATGCCTGGAA CTTTCGGCAGTGCCAGTT	194	59.9 60.0	50.0 55.6	(This work)
*FimH*	*FimH 1* *FimH 2*	GGCTGCGATGTTTCTGCT CCCCAGGTTTTGGCTTTT	105	60.1 59.9	55.6 50	(This work)
*csgD*	*csgD 1* *csgD 2*	CCGTACCGCGACATTGA CGCCTTGCAACCCATT	91	60.2 59.1	58.8 56.2	(This work)
*ariR*	*ariR 1* *ariR 2*	TGTTAGGGCAGGCTGTCA TCGCAACACGATTTCCAG	149	58.9 59.3	55.6 50.0	(This work)
*pgaC*	*pgaC 1* *pgaC 2*	TTGATGGCGATGCGTTATTA GGAATACTCGCCAACCTGAA	153	60.1 60.1	40 50	(This work)
*qseC*	*qseC 1* *qseC 2*	ACCCACGACGGCAGAAT GCCCGTCAGCAAAACCT	88	60.1 59.8	58.8 58.8	(This work)
*RNAr*	*rssA 1* *rssA 2*	AGGTGATCCGCCCGATA CGGCAAAAGTTCGTCCA	130	60.0 59.3	58.8 52.9	(This work)

**Table 5 antibiotics-12-00845-t005:** List of genes and their respective primers for evaluation of the effect of the EO from LOTC on *S. aureus* gene expression during QS communication and biofilm formation.

Gene	Primer	Sequence 5′ to 3′	Product Size (pb)	Tm (°C)	% GC	References
*hla*	*Hla 1* *Hla 2*	GGCCTTATTGGTGCAAATGT CCATATACCGGGTTCCAAGA	176	59.8 59.6	45 50	[72,73,74]
*agrA*	*agrA 1* *agrA 2*	CAACCACAAGTTGTTAAAGCAG TCGTTGTTTGCTTCAGTGATTC	173	57.6 60.3	40.9 40.9	(This work)
*RNAIII*	*RNAIII 1* *RNAIII 2*	CATGGTTATTAAGTTGGGATGGC GAAGGAGTGATTTCAATGGCACA	188	58.31 60.02	43.48 43.48	[75,76]
*icaA*	*icaA 1* *icaA 2*	GAGGTAAAGCCAACGCACTC CCTGTAACCGCACCAAGTTT	151	59.70 59.18	55 50	[77,78]
*icaD*	*icaD 1* *icaD 2*	ACCCAACGCTAAAATCATCG GCGAAAATGCCCATAGTTTC	211	56.99 56.16	45 45	[77,78]
*aur*	*Aur 1* *Aur 2*	ACCGTGTGTTAATTCGTGTGCTA ATGGTCGCACATTCACAAGTTT	65	61.33 59.90	43.49 40.91	[79]
*SarA*	*SarA 1* *SarA 2*	GTAATGAGCATGATGAAAGAACTGT CGTTGTTTGCTTCAGTGATTCG	111	58.44 59.53	36 45.45	[80]
*rbf*	*Rbf 1* *Rbf 2*	AACCACCTAACTGATGTTATAC GACAACTTGACTGTTCTTATTC	156	53.77 53.59	36.36 36.36	[81]
*RNAr*	*Nuc 1* *Nuc 2*	AATATGGACGTGGCTTAGCGT TTGACCTGAATCAGCGTTGTCTT	197	60.38 61.28	47.62 43.48	(This work)

## Data Availability

Data are contained within the article.

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
