# Peer review of "Effect of Essential Oil from Lippia origanoides on the Transcriptional Expression of Genes Related to Quorum Sensing, Biofilm Formation, and Virulence of Escherichia coli and Staphylococcus aureus"

_antibiotics, 2023, doi:10.3390/antibiotics12050845_

Round 1
Reviewer 1 Report (Previous Reviewer 3)
The reviewer congratulates the authors for their efforts to revise the MS according to review reports round 1.
The reviewer kindly encourages the authors to attentively check and revise the references according to MDPI style.
After performing these minor corrections, the reviewer considers that their MS is suitable for publication in Antibiotics MDPI Journal.
The authors are not native English speakers. The reviewer considers that the quality of English language and editing style is good enough for their MS publication in Antibiotics MDPI Journal in the current form, after performing previously mentioned minor corrections.
Author Response
We appreciate the comments of the reviewer. The sugerences were modified in the text.
Reviewer 2 Report (New Reviewer)

Author Response
We appreciate each of the suggestions and we respond to each of them below:
1. “Please go over the manuscript and check for grammatical and spelling errors”.
Response: We appreciate the comments of the reviewer. The mistakes were modified in the text.
2. “Table 2, typo in the E. coli planktonic cell concentration, should be CFU/mL, not UFC/mL”.
Response: We appreciate the comments of the reviewer. The mistakes were modified in the text.
3. “Figure 2 should be repeated with 0.45 mg/mL. This concentration displayed activity with the planktonic bacteria in Figure 1 while 0.37 mg/mL did not show any effect compared to the control. Using 0.37 mg/mL for the biofilm assessment did not yield any useful data since treated biofilm cells showed no reduction at 24 or 48 hrs. Using 0.45 mg/mL or a higher concentration could possible show a reduction in biofilm and add value to the investigation”.
Response: We appreciate the suggestion of reviewer. However, we chose to use the concentration of 0.37 mg/mL since we wanted to verify that the transcriptional effect was directly related to the essential oil and not to a pressure on planktonic growth. As can be seen in figure 1, at a concentration of 0.45 mg/mL the essential oil inhibits 50% of the planktonic cells present, which is reflected in the alteration of biofilm formation. Therefore, with the concentration used in this study we can conclude that the observed transcriptional effects are directly related to the essential oil.
In our previous publication we evaluated different concentrations of essential oil on planktonic and sessile cells, in addition to the effect on the cell membrane of both microorganisms1.
1. Martínez, A., Manrique-Moreno, M., Klaiss-Luna, M. C., Stashenko, E., Zafra, G., & Ortiz, C. (2021). Effect of essential oils on growth inhibition, biofilm formation and membrane integrity of Escherichia coli and Staphylococcus aureus. Antibiotics, 10(12), 1474.
4. “Figure 3, the ladder info and the figure caption need correcting. The term should be “bp” for base-pairs, not “pb”
Response: We appreciate the comments of the reviewer. The mistakes were modified in the text.
Reviewer 3 Report (New Reviewer)
This manuscript was prepared in high quality. I recommend accepting it in present form.
Author Response
We appreciate the feedback from the reviewer. The suggestion have been applied to the main text.
Round 2
Reviewer 2 Report (New Reviewer)
The authors have evaluated the activity of the essential oil (EO) thymol-carvacrol II chemotype (LOTC II) from Lippia origanoides plants. They tested this compound against Escherichia coli and Staphylococcus aureus. The conclusions are supported by the data. The authors have addressed the previous comments and made relevant corrections.
This manuscript is a resubmission of an earlier submission. The following is a list of the peer review reports and author responses from that submission.
Round 1
Reviewer 1 Report
Interest in new antibacterial compounds is readily increasing and natural sources gain interest.
But this paper shows two important pitfalls.
Working on only two strains (which are collection strains subcultured in labs since years) is not a good possibility to analyze the role of recently immerging MDR bacteria.
Author Response
We appreciate the comments and suggestions of the reviewers, which greatly contributed to improving our manuscript.
According to the reviewer’s comment, it is important to mention that previous works have shown that strains that are initially sensitive to antibiotics can be resistant to them when they are wrapped in the biofilm. In consequence, as a preliminary work, we wanted to evaluate whether LOCT II presented an effect on sensitive bacterial models in both their planktonic and biofilm forms (1). In addition, the effect of LOCT II EO in different sensitive and multi-drug resistant microorganisms has been previously proved by our research group (1, 2). Therefore, following the research path outlined, it was essential to give insight into the possible mechanism of action of EO on sensitive reference strains and study models of both Gram-negative and Gram-positive bacteria. For this reason, using this study as a starting point, it will be possible to make a comparison with the studies in progress against multi-resistant bacteria, being able to achieve deeper conclusions about the mechanism of action of the essential oil and the mechanism of protection of microorganisms against it.
1. Martínez, A., Manrique-Moreno, M., Klaiss-Luna, M. C., Stashenko, E., Zafra, G., & Ortiz, C. (2021). Effect of essential oils on growth inhibition, biofilm formation and membrane integrity of Escherichia coli and Staphylococcus aureus. Antibiotics, 10(12), 1474.
2. Cáceres, M., Hidalgo, W., Stashenko, E., Torres, R., & Ortiz, C. (2020). Essential oils of aromatic plants with antibacterial, anti-biofilm and anti-quorum sensing activities against pathogenic bacteria. Antibiotics, 9(4), 147
Reviewer 2 Report
AMR is presently the global burning health issue and the manuscript highlights the plausible role of dietary herbs in ameliorating it and reinforcing the efficacy of conventional antibiotics.
Line 416-18: the definition needs more clarification.
the manuscript can further be improved if the biomarkers for essential oil composition can be added
Author Response
We appreciate the comments and suggestions of the reviewers, which greatly contributed to improving our manuscript. All changes made to the manuscript are detailed below:
- Line 416-18: the definition needs more clarification.
Response: The text has been rephrased as follows:
“The minimum inhibitory concentration of biofilm formation (MIBC) is defined as the lowest concentration of EO to inhibit the formation of biofilm and cause an inhibition of the planktonic growth lower than 10%. “
- The manuscript can further be improved if the biomarkers for essential oil composition can be added.
Response: We appreciate the suggestions of the Reviewer.
"The major biomarkers for essential oils composition have been previously added in Table 1, highlighting thymol and carvacrol compounds. See lines 102-107 from the new corrected version: “Five major components present in L. origanoides EO were identified by GC-MS analyses [28] (See Table 1). The percentage of major biomarkers are as follows: oxygenated compounds 51.5%, sesquiterpenes 6.3% and monoterpenes 6.4%, respectively. Among them, the major biomarkers thymol (32.7%) and carvacrol (18.8%) have been previously proven as promising antimicrobial compounds against antibiotic-resistant bacteria [29]".
Reviewer 3 Report
In their MS entitled "Effect of EO from Lippia origanoides (Verbenaceae) on the transcriptional expression of genes related to quorum sensing, biofilm formation, and virulence of Escherichia coli and Staphylococcus aureus," the authors aimed a complex analysis of the mechanisms that underly the antibacterial effect of L. origanoides essential oil against both MDR bacteria.
The study is extensive and well-designed. The Introduction offers a suitable background for the aim of their work. The figures are relevant to the entire study, and the results and discussion are detailed enough to support the conclusions.
The available comments are displayed below:
1. The reviewer suggests avoiding the abbreviation "EO" in the MS title. If the title appears too long, they could remove the family name (Verbenaceae), mentioning it only in the MS text.
2. Lines 25-27 - Please, reformulate this phrase for better understanding, and check and correct the words included in parentheses. Maybe it is better to introduce the abbreviation LOTC II EO in this phrase to simplify the following ideas about L. origanoides EO.
3. Line 40-42 - The authors are encouraged to reformulate the phrase for better understanding.
4. Lines 34, 68: The authors are invited to show on the first mention in the MS text the entire name of PIA/PNG - PIA (polysaccharide intercellular adhesin) and PNG (poly N-acetylglucosamine); in most publications, the abbreviation of the second term is PNAG. Moreover, they should use the same abbreviation in the entire MS text for uniformity because, in Discussion (lines 213, 214, 230, 272), PIA/PNAG is used.
5. Line 99: The authors are encouraged to continue the aim of their study presentation, highlighting its novelty.
6. Line 106. QUI VOY?
7. Line 113, 121, 140, 162, 191, 200: A single mention of abbreviation is sufficient for the entire MS text. The authors are encouraged to mention the abbreviations' significance in Table footers and Figure captions.
8. Figures 2-4. For a better visibility, the authors should augment the smallest characters (letters and numbers).
9. The authors mentioned in the Data analysis (line 483) that all experiments were performed in triplicate. They are encouraged not to repeat it (as in lines 122-123 and 136); however, in Results, they could register the obtained values as a mean ± SD (SD=standard deviation).
10. The reviewer invites the authors to check and correct the following mention: All experiments were performed in triplicate and ANOVA (****p=<0.0001, 136 ***p=<0.0002) analysis (lines 136-137). Moreover, the reviewer suggests that the place of this mention is not in the Figure 1 caption; it is suitable for the Figures 4 and 5 captions. In both Figures 4 and 5 captions, the authors should add the signification of ** and "ns" notations; maybe on the horizontal axis, "evaluated genes" (as in the MS text) is better.
11. Line 154: "Extraction of total RNA extracted..." - The repetition should be avoided.
12. Line 157: maybe "ratio" is better
13. Line 381. The authors are encouraged to entitle the first subsection of Materials and Methods, 4.1. Materials, where they can display all materials used in their study: plant material, bacteria, culture media, reagents, and chemicals, with their provenance (Producer name, City, State - for UK and USA - and country). Thus, the description of the methods will be substantially clear and concise.
14. Lines 166, 397-398 - Bacteria names with italics
15. The authors are encouraged to edit the references according to MDPI instructions.
Author Response
We appreciate the comments and suggestions of the reviewers, which greatly contributed to improving our manuscript. All changes made to the manuscript are detailed in the attachment.

Round 2
Reviewer 1 Report
No substantial changes occurred in the new version!